# Single platinum atoms embedded in nanoporous cobalt selenide as electrocatalyst for accelerating hydrogen evolution reaction

Kang Jiang[1,7], Boyang Liu[2,7], Min Luo[3,7], Shoucong Ning [4], Ming Peng[1], Yang Zhao[1], Ying-Rui Lu[5], Ting-Shan Chan[5], Frank M.F. de Groot[6] & Yongwen Tan [1]

Designing efficient electrocatalysts for hydrogen evolution reaction is significant for renewable and sustainable energy conversion. Here, we report single-atom platinum decorated nanoporous $Co_{0.85}Se$ ($Pt/np-Co_{0.85}Se$) as efficient electrocatalysts for hydrogen evolution. The achieved $Pt/np-Co_{0.85}Se$ shows high catalytic performance with a near-zero onset overpotential, a low Tafel slope of 35 mV dec$^{-1}$, and a high turnover frequency of 3.93 s$^{-1}$ at −100 mV in neutral media, outperforming commercial Pt/C catalyst and other reported transition-metal-based compounds. Operando X-ray absorption spectroscopy studies combined with density functional theory calculations indicate that single-atom platinum in $Pt/np-Co_{0.85}Se$ not only can optimize surface states of $Co_{0.85}Se$ active centers under realistic working conditions, but also can significantly reduce energy barriers of water dissociation and improve adsorption/desorption behavior of hydrogen, which synergistically promote thermodynamics and kinetics. This work opens up further opportunities for local electronic structures tuning of electrocatalysts to effectively manipulate its catalytic properties by an atomic-level engineering strategy.

[1] College of Materials Science and Engineering, Hunan University, 410082 Changsha, Hunan, China. [2] Department of Physics, AlbaNova University Center, Stockholm University, SE-10691 Stockholm, Sweden. [3] Department of Physics, Shanghai Polytechnic University, 201209 Shanghai, China. [4] Department of Materials Science and Engineering, National University of Singapore, 9 Engineering Drive 1, Singapore 117575, Singapore. [5] National Synchrotron Radiation Research Center, Hsinchu 300, Taiwan. [6] Inorganic Chemistry & Catalysis, Debye Institute for Nanomaterials Science, Utrecht University, Universiteitsweg 99, 3584 CG Utrecht, The Netherlands. [7] These authors contributed equally: Kang Jiang, Boyang Liu, Min Luo. Correspondence and requests for materials should be addressed to Y.T. (email: tanyw@hnu.edu.cn)

Electrocatalytic water splitting to generate hydrogen is widely regarded as an efficient and prospective sustainable energy technology[1]. Furthermore, electrochemically generated $H_2$ may be used as feedstock in industrially relevant reactions like Fischer-Tropsch and Haber-Bosch, which are currently dependent on fossil-fuel derived $H_2$ (i.e., derived from steam reforming of natural gas). Highly efficient catalysts with low overpotential and fast kinetics in the hydrogen evolution reaction (HER) are critical toward large-scale hydrogen production. Pt is generally considered as the most active catalyst for HER, but their scarcity and high cost greatly hamper its large-scale applications[2]. For cost-efficiency, numerous non-precious materials have been found to have excellent HER catalytic performance, including metal oxides or hydroxides[3–5], metal nitrides[6], metal chalcogenides[7–16], and metal phosphides[17–23]. Among them, the non-stoichiometric cobalt selenides ($Co_{0.85}Se$) are one type of promising catalysts for HER applications[24–27]. However, ever-reported catalysts based on non-noble metal suffer from sluggish catalytic kinetics and poor stability, thus calling for further exploitation of highly efficient HER catalysts through material structure innovation.

Recently, many researches have suggested that supported Pt nanostructures are typically used to promote catalytic activity by advantageous local catalyst–Pt interfacial interactions[3–5,9]. Given the noble nature of Pt, reducing the nanostructures to atomically distributed Pt centers supported on catalysts could significantly decrease Pt usages and maximize atom efficiency[28,29]. In particular, significant progresses have been made on the constructing of atomic-scale Pt on various supports, such as metal[30], metal oxides[31], metal phosphides[32], two-dimensional $MoS_2$[11,14], carbon nanotubes[33], graphene[28,34], MXene[35], etc. Some of these works have demonstrated that the single metal atoms modified catalysts are significantly higher catalytic activity than non-modified ones towards HER due to strong metal-support interactions (SMSI)[31,36]. However, at present, direct insights into how single-atom metal is beneficial to promote HER activity of current catalysts are rarely available under realistic reaction conditions, especially for cobalt selenides. So from feasible and representative perspectives, developing single Pt atoms modified cobalt selenides catalyst is key to fundamentally understanding the synergistic effect of single Pt atoms and cobalt selenides that accounts for HER enhancement.

Herein, we construct single-atom Pt supported on nanoporous cobalt selenide (denoted as Pt/np-$Co_{0.85}Se$) catalyst as a model to assess the HER activity and understand the activity origins at atomic-level. The Pt/np-$Co_{0.85}Se$ catalyst with an ultralow single-atom Pt loading while possess a near-zero onset overpotential, a low Tafel slope, and a high turnover frequency (TOF), outperforming commercial Pt/C catalyst and other reported transition-metal-based compounds in neutral electrolyte. Operando X-ray absorption spectroscopy (XAS) studies combined with density functional theory (DFT) calculations reveal that single Pt atoms strongly induce the charge redistribution at the interface region of Pt/np-$Co_{0.85}Se$ and significantly promote water dissociation process, while improve adsorption/desorption behavior of H, further facilitating the HER kinetics. Thus, the synergy between atomic-level Pt and np-$Co_{0.85}Se$ is mainly responsible for excellent catalytic activity of Pt/np-$Co_{0.85}Se$ in HER processes. The superior catalytic performance of the Pt/np-$Co_{0.85}Se$ catalyst highlights the importance of atomic-level engineering strategy for electronic structure tuning of electrocatalysts to effectively manipulate theirs catalytic properties.

## Results

**Material synthesis and characterization.** The np-$Co_{0.85}Se$ catalyst was prepared by an electrochemically selective etching method (Methods and Supplementary Figs. 1–4)[37]. Inspired by the potential cycling method of depositing Pt atom on working electrode[32,33], electrochemical vacancy manufacturing and Pt atom embedding were conducted by cyclic voltammetry (CV) using Pt foil as counter electrode in a three-electrode cell containing 0.5 M $H_2SO_4$ (Fig. 1a, Methods). During the cyclic process, slight Co atoms dissolved from the np-$Co_{0.85}Se$ to form Co vacancies, thus providing anchor sites for the embedding of Pt atoms and improving the HER performance (Supplementary Note 1 and Supplementary Figs. 5 and 6)[38,39]. The bicontinuous nanoporous structure with a high surface area facilitates Pt ions' diffusing into Co vacancies of nano-sized $Co_{0.85}Se$ ligaments, forming the homogeneous single-atom Pt doped np-$Co_{0.85}Se$ (Supplementary Fig. 7). CV measurements under non-catalytic conditions were used to detect the feature voltammetric response of Pt[40]. Regions of interest include the Pt-H adsorption/desorption peaks and Pt-O formation/reduction peaks. However, these characteristic peaks of Pt cannot be observed on Pt/np-$Co_{0.85}Se$, indicative of ultralow Pt loading (Supplementary Fig. 8)[33]. X-ray diffraction (XRD) pattern of Pt/np-$Co_{0.85}Se$ displays a similar crystal structure with that of hexagonal $Co_{0.85}Se$ (JPCDS card no. 52–1008) excepting for a slight shift (Fig. 1b and Supplementary Fig. 9), indicating that the atomic-level Pt dopant did not form crystalline segregated bi-phases, but was well-incorporated into the $Co_{0.85}Se$ crystal lattice. Scanning electron microscope (SEM) (Fig. 1c) and transmission electron microscopy (TEM) (Supplementary Fig. 10) characterizations confirmed that Pt/np-$Co_{0.85}Se$ remains the porous morphology of np-$Co_{0.85}Se$ precursor after surface doping, and no Pt nanoparticles are observed from the surface of np-$Co_{0.85}Se$. High-angle annular dark-field scanning TEM (HAADF-STEM) measurements were adopted to directly observe the presence of Pt on the surface of np-$Co_{0.85}Se$. As Fig. 1d, e shown, single-atom Pt appearing as bright spots can be found to be well dispersed in the lattice of $Co_{0.85}Se$, confirming the formation of single-atom dispersed catalyst. Similarly, the loosely interatomic distances and the different intensity of line profiles also indicate the isolated Pt atoms (Fig. 1f). The energy-dispersive X-ray (EDX) spectroscopy elemental analysis further demonstrate the homogenous distribution of Co, Se, and Pt throughout the ligaments surface in Pt/np-$Co_{0.85}Se$ (Fig. 1g), conforming the uniform dispersion of single-atom Pt on np-$Co_{0.85}Se$. ~1.03 wt% Pt loading was determined for the Pt/np-$Co_{0.85}Se$ catalyst, as ascertained by inductively coupled plasma optical emission spectrometry (ICP-OES) results. By tuning the CV cycles (Supplementary Fig. 5), lower mass loading catalyst with nonuniform distribution of single Pt atoms (denoted as $Pt_S$/np-$Co_{0.85}Se$, 3000 CV cycles) and higher mass loading catalyst with Pt nanoparticles (denoted as $Pt_N$/np-$Co_{0.85}Se$, 6000 CV cycles) were synthesized. This variation in size from single atoms, nanoclusters to nanoparticles is verified by HAADF-STEM measurements (Supplementary Fig. 11).

X-ray photoelectron spectroscopy (XPS) was performed to investigate the chemical composition and binding status of the catalysts. Figure 2a shows the Pt 4f core level spectra of Pt/C and Pt/np-$Co_{0.85}Se$. Pt $4f_{7/2}$ and $4f_{5/2}$ orbitals of the Pt/C are observed at binding energies of 71.2 eV and 74.5 eV, respectively, indicative of $Pt^0$[32]. However, the two Pt 4 f orbitals of the Pt/np-$Co_{0.85}Se$ are located at 71.8 eV and 75.1 eV, which confirm Pt atoms with partially positive charge ($Pt^{\delta+}$) owing to the electronic interaction between single Pt atoms and np-$Co_{0.85}Se$[41,42]. In the Co 2p region, enhanced oxidation degree of the surface Co cations after atomic-level Pt doping causes a positive energy shift in Co XPS (Supplementary Fig. 12a). The fitted ratios of $Co^{3+}$ to $Co^{2+}$ of Pt/np-$Co_{0.85}Se$ (2.27) are higher than that of pristine np-$Co_{0.85}Se$ (1.02), implying the more $Co^{2+}$ vacancies formed during Pt doping according to the charge neutrality[43]. In the Se 3d spectra,

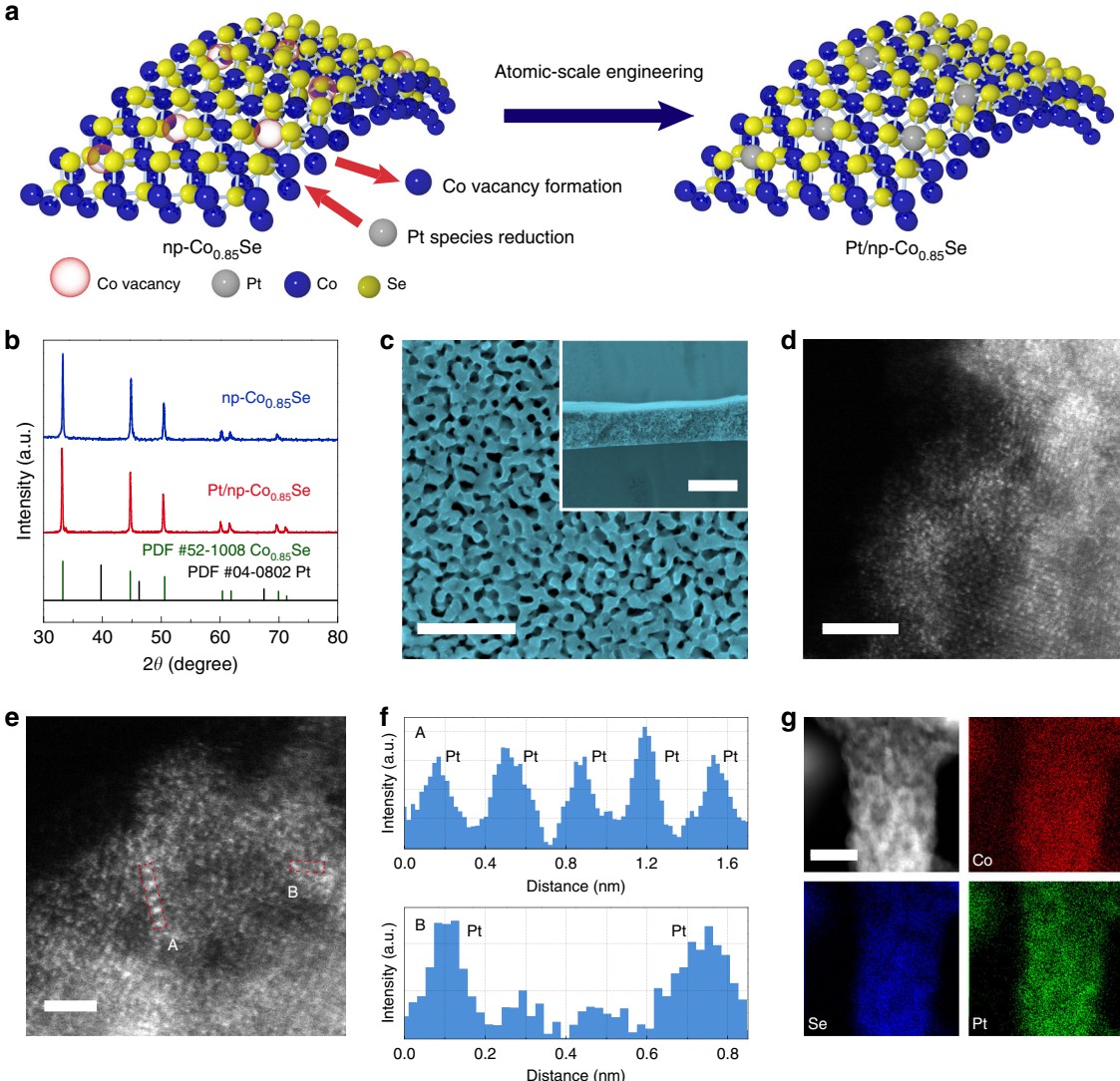

**Fig. 1** Fabrication and structural characterization of Pt/np-Co$_{0.85}$Se. **a** Schematic illustration of the fabrication procedure. **b** X-ray diffraction (XRD) patterns of np-Co$_{0.85}$Se and Pt/np-Co$_{0.85}$Se. **c** SEM image of Pt/np-Co$_{0.85}$Se. Inset shows the microstructure of cross-section of Pt/np-Co$_{0.85}$Se. **d**, **e** HAADF-STEM images of Pt/np-Co$_{0.85}$Se. **f** Line-scanning intensity profile obtained from the area highlighted with red rectangles in regions A and B in **e**. **g** The STEM-EDX elemental mapping of Pt/np-Co$_{0.85}$Se. Scale bars: **c** 500 nm, inset: 20 μm. **d** 2 nm. **e** 1 nm. **g** 10 nm

the peaks of Se 3d for Pt/np-Co$_{0.85}$Se show a positive energy shift compared to that of np-Co$_{0.85}$Se, which is ascribed to the doping of Pt in lattice resulting in the increased electron density (Supplementary Fig. 12b)[44].

X-ray absorption spectroscopy (XAS) was used to investigate the electronic and local structure of catalysts[45]. Figure 2b, c exhibit Pt L$_3$-edge X-ray absorption near-edge structure (XANES) (Fig. 2b) and Fourier transform extended X-ray absorption fine structure (FT-EXAFS) (Fig. 2c) spectra, together with the Pt foil, commercial Pt/C, PtO$_2$ as a comparison. The XANES spectra show that the white-line intensity of Pt/np-Co$_{0.85}$Se is obviously higher than that of Pt foil and commercial Pt/C, confirming the positive valence state (0.8, inset of Fig. 2b) of Pt atoms, which could be attributed to the electron transfer from Pt to Se in Pt-Se bonds of Pt/np-Co$_{0.85}$Se[14]. FT-EXAFS spectra in Fig. 2c show a conspicuous peak at 2.03 Å from the Pt-Se contribution, indicating the single-atom nature of Pt in Pt/np-Co$_{0.85}$Se. Figure 2d, e exhibit the Se K-edge XANES and FT-EXAFS spectra of np-Co$_{0.85}$Se and Pt/np-Co$_{0.85}$Se, respectively. The FT-EXAFS spectrum (Fig. 2e) of np-Co$_{0.85}$Se exhibits a coordination

peaks of Se-Co at 2.09 Å, ~0.06 Å longer than that of Se-Pt, which suggests the equal radial distance of Se-Co and Se-Pt shell[16]. Moreover, the enhanced intensity of Se-Co (Se-Pt) shell in the FT-EXAFS signals indicates that Pt atoms occupy Co vacancies in the lattice of np-Co$_{0.85}$Se, resulting in more the coordination number of Se atoms. The Co K-edge XANES spectra of np-Co$_{0.85}$Se and Pt/np-Co$_{0.85}$Se show similar but slightly different adsorption features (Fig. 2f). The higher intensity white-line of Pt/np-Co$_{0.85}$Se compared to np-Co$_{0.85}$Se implies the local atomic arrangement of Co atom caused by single-atom Pt doping. This phenomenon is further definitely verified by the FT-EXAFS spectra (Fig. 2g). A prominent peak is observed at 2.09 Å, corroborating the Co-Se scattering contribution. The peak intensity decreases after the introduction of the single-atom Pt in np-Co$_{0.85}$Se, which might originate from the formation of Se vacancies for the partial rearrangement of Co atoms[11]. More importantly, the peak from the Co-Se contribution shifts to a higher value by 0.04 Å compared to that of np-Co$_{0.85}$Se, which may originate from the structural lattice distortion caused by the substitutional doping of Pt.

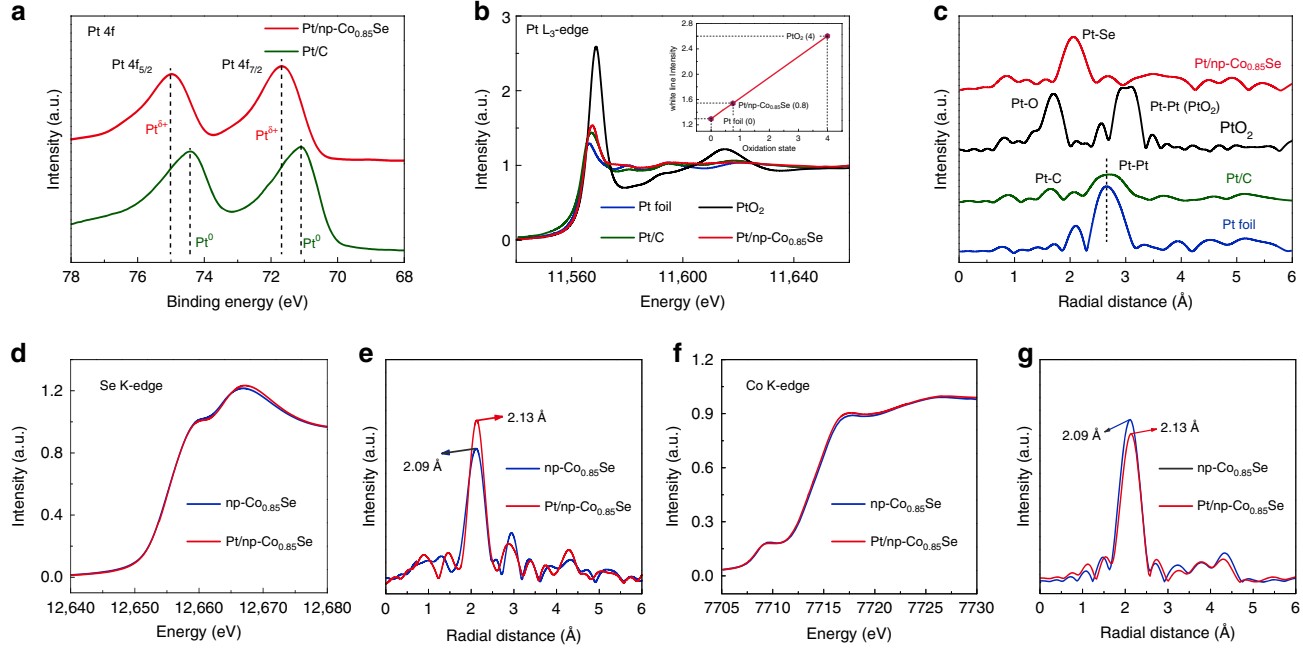

**Fig. 2** X-ray photoelectron spectroscopy (XPS) and X-ray absorption spectroscopy (XAS) characterizations. **a** XPS spectra of Pt/np-$Co_{0.85}$Se and commercial Pt/C in Pt 4f regions. **b** The normalized XANES at the Pt $L_3$-edge of Pt foil, commercial Pt/C, $PtO_2$, and Pt/np-$Co_{0.85}$Se. The inset shows the average oxidation state of Pt in Pt/np-$Co_{0.85}$Se. **c** Corresponding FT-EXAFS spectra from **b**. **d** The normalized XANES spectra at the Se K-edge of np-$Co_{0.85}$Se and Pt/np-$Co_{0.85}$Se. **e** Corresponding FT-EXAFS spectra from **d**. **f** The normalized XANES spectra at the Co K-edge of np-$Co_{0.85}$Se and Pt/np-$Co_{0.85}$Se. **g** Corresponding FT-EXAFS spectra from **f**

**Electrochemical analysis.** The HER activities of Pt/np-$Co_{0.85}$Se was evaluated in a three-electrode system containing 1.0 M phosphate buffer solutions (PBS, pH = 7.0). Figure 3a exhibits the polarization curves of np-$Co_{0.85}$Se and Pt/np-$Co_{0.85}$Se, together with a commercial Pt/C as the benchmark. Although comparable HER activity at low applied potentials, Pt/np-$Co_{0.85}$Se can far surpass commercial Pt/C at high overpotentials (greater than −19 mV), presumably owing to its bicontinuous nanoporous structure that enables better mass-transfer process. The potential required to reach an HER current density ($j$) of −10 mA cm$^{-2}$ is a key HER performance metric. Only −55 mV vs. reversible hydrogen electrode (RHE) overpotential was required for Pt/np-$Co_{0.85}$Se to reach 10 mA cm$^{-2}$. In comparison, the np-$Co_{0.85}$Se requests the overpotential of −264 mV for $j = -10$ mA cm$^{-2}$ electrode current. The tafel slope presented in Fig. 3b gives a small Tafel slope of 35 mV per decade (mV dec$^{-1}$) for Pt/np-$Co_{0.85}$Se, lower than that of Pt/C (37 mV dec$^{-1}$) and np-$Co_{0.85}$Se (90 mV dec$^{-1}$), revealing fast HER kinetics derived from the advantage of introducing Pt single atoms. Additionally, the Pt/np-$Co_{0.85}$Se electrode shows the onset potential (potential required to reach −1 mA cm$^{-2}$) for $H_2$ evolution at −12 mV (Fig. 3c), which can only be observed for commercial Pt/C catalyst. Whereas the onset potential is shifted substantially negative for np-$Co_{0.85}$Se catalyst. Furthermore, the mass activity of HER for Pt/np-$Co_{0.85}$Se at an overpotential of −100 mV is 1.32 A mg$^{-1}$ by normalizing to the Pt loading, which is 11 times greater than that of the commercial HER catalyst (10 wt% Pt/C, 0.12 A mg$^{-1}$) (Fig. 3d and Supplementary Note 2) and also higher than that of Pt-based catalysts reported recently (Supplementary Table 1)[32,46]. This result indicates that single-atom Pt anchored on the np-$Co_{0.85}$Se can maximize the catalytic activity, decreasing the cost of HER catalysts. The above merits of the Pt/np-$Co_{0.85}$Se, including low overpotential and Tafel slope, are superior to commercial Pt/C and most previously reported catalysts in the neutral solution (Fig. 3e and

Supplementary Table 2)[12,19,21,22,47]. The TOF of Pt/np-$Co_{0.85}$Se at −100 mV vs. RHE were calculated to be 3.93 s$^{-1}$ (Fig. 3f and Supplementary Note 3), which is better than that of np-$Co_{0.85}$Se (0.17 s$^{-1}$) and most reported catalysts (Supplementary Table 3)[23,48–51]. Afterwards, gas chromatography was introduced to analyze the $H_2$ production, which shows that the Faraday efficiency of Pt/np-$Co_{0.85}$Se is close to 100% under different applied potentials (Fig. 3g, h).

Electrochemical impedance spectroscopy (EIS) analysis further evidences that the introduction of atomic-level Pt in np-$Co_{0.85}$Se generates low internal resistance and rapid charge transfer behavior for a low onset potential and fast HER kinetics (Supplementary Fig. 13). Further, double-layer capacitance ($C_{dl}$), which was used as indicator of the effective electrochemically active surface area, was examined for studied catalysts. Our results reveal a considerably larger $C_{dl}$ of Pt/np-$Co_{0.85}$Se (57.2 mF cm$^{-2}$) compared with np-$Co_{0.85}$Se (34.1 mF cm$^{-2}$), implying more accessible active sites constructed on Pt/np-$Co_{0.85}$Se catalyst (Supplementary Fig. 14). Beside activity, we performed an accelerated cyclic voltammetry cycling test to evidence this catalytic robustness with negligible shift of polarization curves after 3000 cycles (Fig. 3i). A long-term stability testing on Pt/np-$Co_{0.85}$Se catalyst by means of chronoamperometry (j ~ t), showing negligible current decay over 40 h at a constant voltage operation (Fig. 3j). The SEM characterizations of the Pt/np-$Co_{0.85}$Se electrode also confirm that detectable morphology changes cannot be seen after long-term HER operation (Supplementary Fig. 15a, b). The HAADF-STEM images (Supplementary Fig. 15c, d) and XAS results (Supplementary Fig. 16) confirmed that the single-atom Pt remained after the long-term HER operation, which indicate that the single-atom Pt possesses excellent stability. In addition to neutral electrolyte, Pt/np-$Co_{0.85}$Se also has high activity and stability toward HER in acidic and basic electrolytes (Supplementary Figs. 17–18 and Supplementary Table 1).

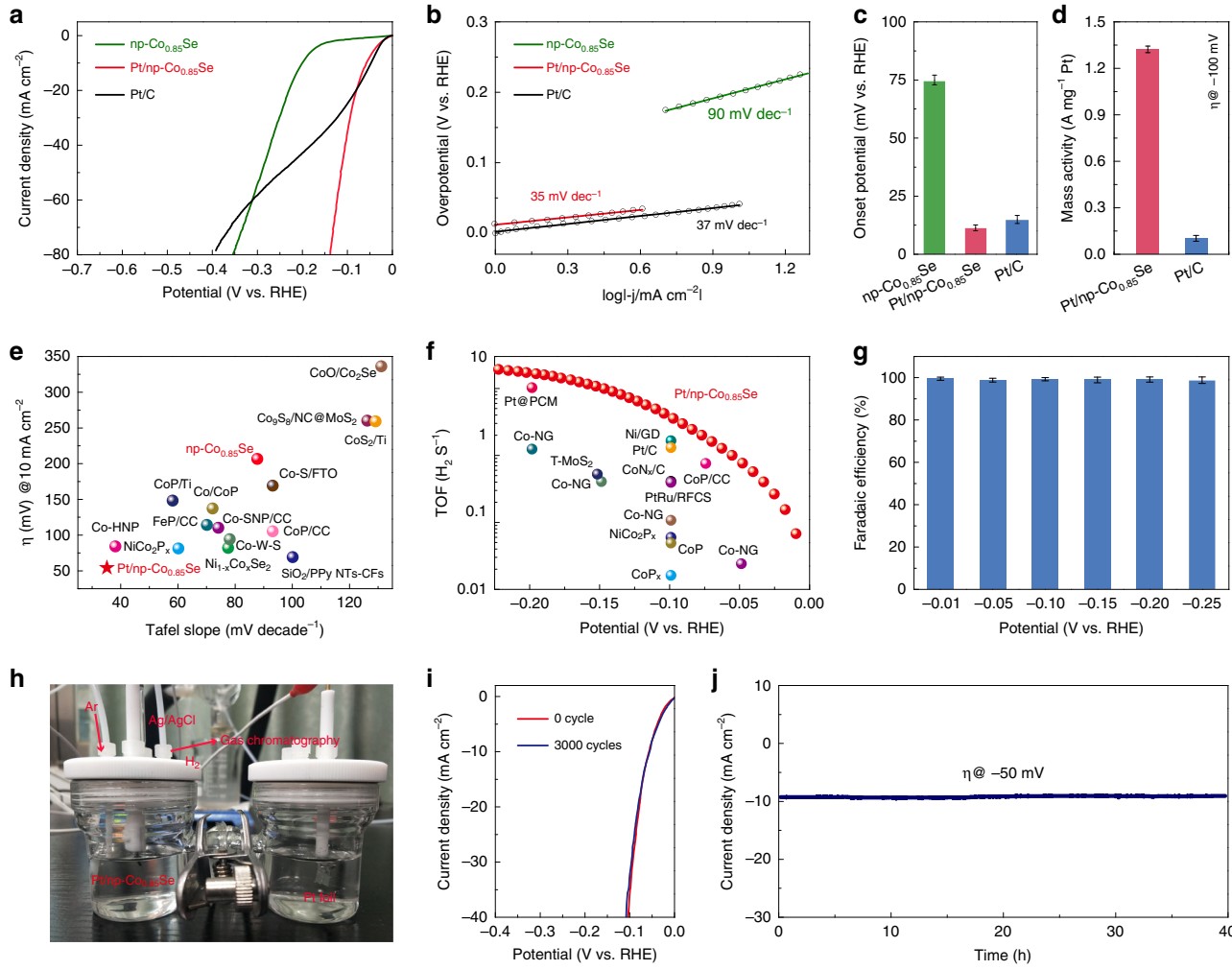

**Fig. 3** Electrochemical hydrogen evolution reaction (HER) performance. **a** HER polarization curves of np-Co$_{0.85}$Se, Pt/np-Co$_{0.85}$Se and Pt/C. **b** Corresponding to Tafel plots of the presented data in **a**. **c** Corresponding to onset potential at −1 mA cm$^{−2}$ of the presented data in **a**. **d** The mass activity of Pt/np-Co$_{0.85}$Se and state-of-the-art Pt/C. **e** Comparison of merit with respect to both kinetics (Tafel slope) and activity (the overpotential required to achieve −10 mA cm$^{−2}$), with references all measured in neutral medium. **f** TOF values of Pt/np-Co$_{0.85}$Se (red dot), together with previous reported HER electrocatalysts at −100 mV vs. RHE. **g** Faradaic efficiency of Pt/np-Co$_{0.85}$Se at different applied potentials. **h** The detail of hydrogen Faradaic efficiency measurement. **i** Accelerated HER polarization curves of Pt/np-Co$_{0.85}$Se. **j** Current density vs. time (i–t) curves of Pt/np-Co$_{0.85}$Se recorded for 40 h s at −50 mV vs. RHE

**HER enhancement mechanism**. To gain insights on the origins of the high activities of Pt/np-Co$_{0.85}$Se in the neutral electrolyte, The in situ and operando Co K-edge XANES and FT-EXAFS spectra were measured under HER working conditions to probe the electronic structure and local atomic environment changes of np-Co$_{0.85}$Se and Pt/np-Co$_{0.85}$Se with a homemade operando cell (Supplementary Note 4 and Fig. 19). During the measurements, the working electrode potential was first increased in steps from the open circuit voltage (OCV, ~0.75 V vs. RHE) to −0.2 V vs. RHE, and then decreased back to OCV. The XAS spectra were recorded under each potential for at least 1 h. Figure 4a, b show in situ and operando Co K-edge XANES spectra of np-Co$_{0.85}$Se and Pt/np-Co$_{0.85}$Se, respectively. In Fig. 4a, with increase of the operated bias voltages from the OCV to −0.2 V vs. RHE, the absorption onset of np-Co$_{0.85}$Se shows no distinct change. While that of the Pt/np-Co$_{0.85}$Se (Fig. 4b) shows the slight shift toward the higher energy, which is more obviously indicated by the first-order derivatives of the XANES spectra as shown in Fig. 4c, d. The different behavior suggests the role of Pt and Co$_{0.85}$Se electronic interactions on facilitating electron transfer from Co to Se atoms during HER. Moreover, similar behaviors of the Co-Se

shell radial distance are also observed from the FT-EXAFS spectra shown in Fig. 4e, f. The shrinking of the radial distance of the Co-Se shell with the applied bias is observed on Pt/np-Co$_{0.85}$Se (Fig. 4f), while not on np-Co$_{0.85}$Se (Fig. 4e), which further suggests the increase of electron intensities on Pt/Co$_{0.85}$Se[20]. Since water as a typical polar molecule consist of two H atoms carrying positive charge and an oxygen atom carrying negative charge, its oxygen atom could be easily captured by positively charged Co atom in Pt/np-Co$_{0.85}$Se, thus completing the adsorption and activation of water molecule. Moreover, the enhanced intensity of Co-OH shell during HER for Pt/np-Co$_{0.85}$Se indicates the more OH$_{ads}$ adsorbed on Co atoms in Volmer reaction of HER process compared to that of Co$_{0.85}$Se[52]. The above results indicate that Pt dopant might optimize the electronic structure of the surrounding Co atoms during HER, which further accelerates the H$_2$O adsorption and dissociation processes. According to the in situ and operando XAS results of Pt/np-Co$_{0.85}$Se during HER, H$_2$O molecules are selectively adsorbed or bound on the Co sites at the early stages of the HER potential region (Step I depicted in Fig. 4g). Then, the H$_2$O molecules in neutral media adsorb electrons that can be dissociated into intermediate H$_{ads}$ and

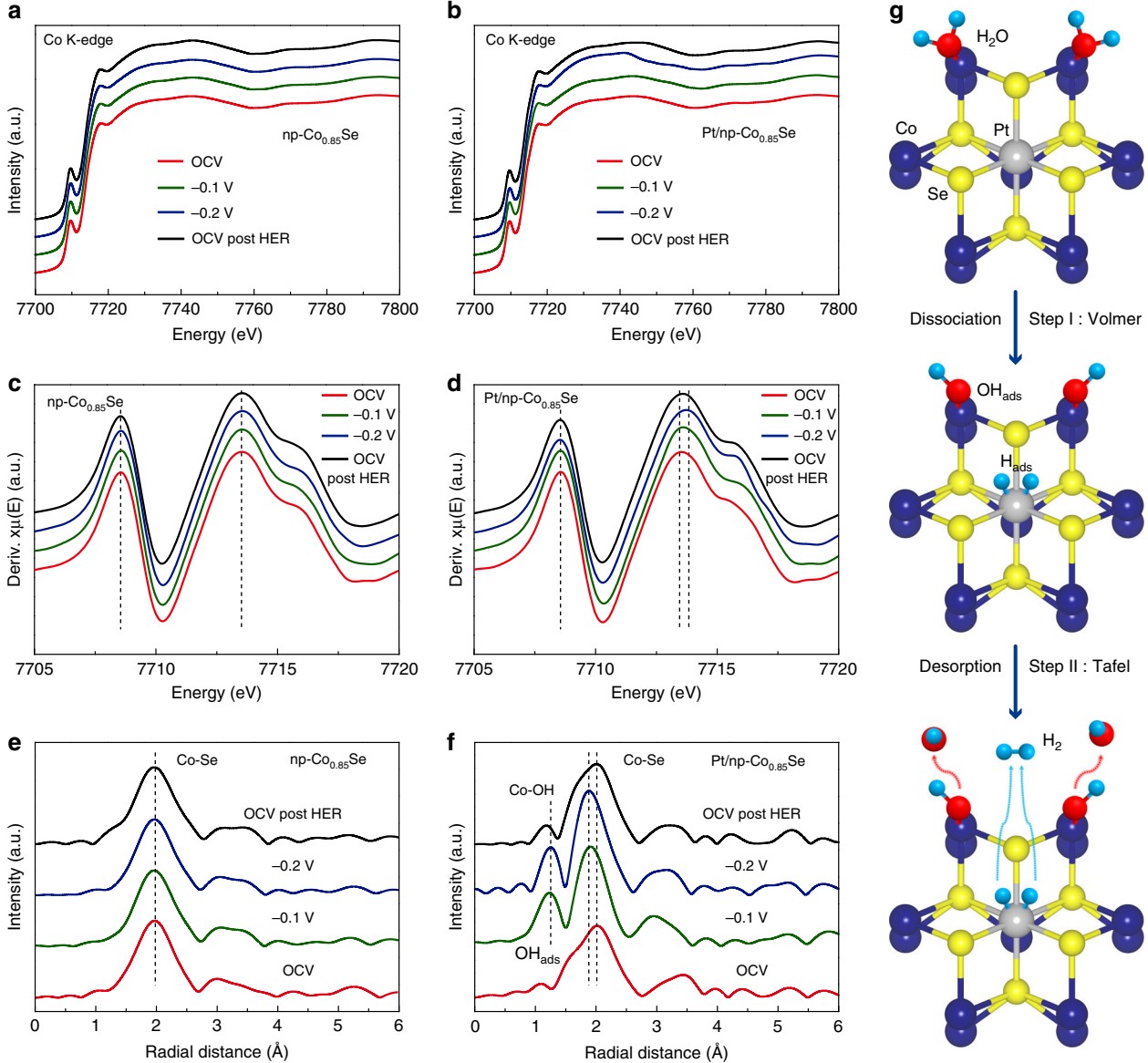

**Fig. 4** In situ and operando X-ray absorption spectroscopy (XAS) characterizations. **a**, **b** Co K-edge XANES of **a** np-$Co_{0.85}$Se and **b** Pt/np-$Co_{0.85}$Se from OCV to −0.2 V (vs. RHE) in 1.0 M PBS. **c**, **d** First-order derivatives of the XANES spectra of **c** np-$Co_{0.85}$Se and **d** Pt/np-$Co_{0.85}$Se. **e**, **f** FT-EXAFS spectra of **e** np-$Co_{0.85}$Se and **f** Pt/np-$Co_{0.85}$Se. **g** Schematic illustration of the hydrogen evolution reaction (HER) mechanism determined by in situ and operando XAS analysis of Pt/np-$Co_{0.85}$Se in neutral media

$OH_{ads}$ by Co sites through the Volmer step. Simultaneously, the generated $H_{ads}$ could be adsorbed on a nearby empty Co site or Pt site and further be converted into $H_2$ readily through the Tafel step (Step II shown in Fig. 4g).

Density functional theory (DFT) calculations were further constructed to examine how the individual components of Pt/$Co_{0.85}$Se cooperate synergistically to enhance the neutral HER activity (Supplementary Note 5 and Figs. 20–22). The charge density difference images (Fig. 5a) reveal a strong charge redistribution at Pt-bonding region (red arrows) after the presence of single-atom Pt in Pt/$Co_{0.85}$Se, which promotes a significant increase in the internal electron concentration of the system, thus enhancing the HER performance. Moreover, the projected density of states (PDOS) results reveal that the Pt dopant gives rise to some new hybridized electronic states in Pt/$Co_{0.85}$Se (Fig. 5b), which could be ascribed to the hybridization between Pt (5d orbitals) and Se atoms. Specifically, the comparison between the PDOS of the $Co_{0.85}$Se and Pt/$Co_{0.85}$Se

reveals that the change near the Fermi level is mainly contributed by the Co 3d orbitals. This indicates that the Pt dopant could effectively optimize the d-electron domination of Co atoms, thus leading to enhanced catalytic activity. This result is consistent with the aforementioned XAS results (Fig. 2f, g). The DFT calculations were also employed to understand the kinetic energy barriers of the HER process. As shown in Fig. 5c, $Co_{0.85}$Se has a very large water dissociation energy barrier ($\Delta G(H_2O) = 0.891$ eV), indicating an extremely sluggish Volmer process. In contrast, the $\Delta G(H_2O)$ of the Pt/$Co_{0.85}$Se dramatically decrease to only 0.491 eV, even lower than that of Pt (111) (0.563 eV), suggesting that the sluggish Volmer process can be greatly accelerated after the introduction of atomic-level Pt dopant[53]. Additionally, the calculated hydrogen adsorption free energy ($\Delta G_H$) in Fig. 5d shows a $\Delta G_H$ value of −0.083 eV for Pt/$Co_{0.85}$Se at Co sites and a $\Delta G_H$ value of −0.079 eV for Pt/$Co_{0.85}$Se at Pt sites, which are comparable to that of Pt (111) (Supplementary Fig. 22). These results indicate that the Co sites are the active site for HER on

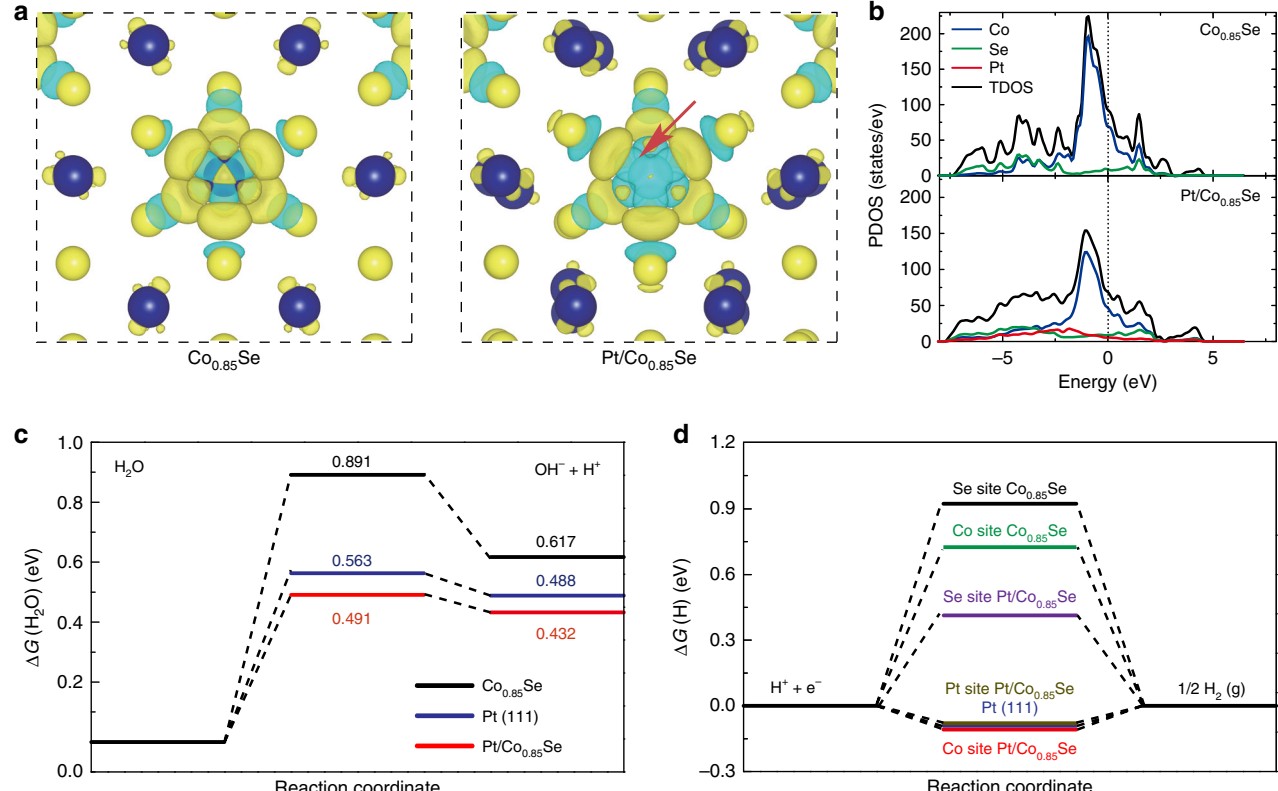

**Fig. 5** Density functional theory (DFT) calculations. **a** Calculated spin density distribution in the $Co_{0.85}Se$ supercell before and after Pt atoms doping. The blue and yellow balls refer to Co and Se atoms. Yellow and cyan isosurfaces represent positive and negative spin densities (0.005 e/$Å^3$), respectively. **b** Calculated DOS of $Co_{0.85}Se$ and $Pt/Co_{0.85}Se$. **c** Calculated adsorption free energy diagrams for the Volmer step on the as-built np-$Co_{0.85}Se$, Pt (111) and $Pt/Co_{0.85}Se$ models. **d** Free energy diagrams for hydrogen adsorption at different active sites of $Co_{0.85}Se$ (004), Pt (111), and $Pt/Co_{0.85}Se$ (004)

$Pt/Co_{0.85}Se$, and the proceeding of adsorption/desorption of H on $Pt/Co_{0.85}Se$ is easier than on $Co_{0.85}Se$. Therefore, the introduction of Pt single-atom in Pt/np-$Co_{0.85}Se$ not only optimizes surface states of $Co_{0.85}Se$ active centers and reduces energy barriers of dissociated water molecules, but also significantly improves adsorption/desorption behavior of H, which synergistically promote the HER thermodynamics and kinetics[10].

## Discussion

In summary, we have demonstrated that atomic engineering of np-$Co_{0.85}Se$ by single Pt atoms doping is an effective approach to produce highly active and robust electrocatalysts for hydrogen evolution in aqueous media. The achieved Pt/np-$Co_{0.85}Se$ shows high HER activity with a near-zero onset overpotential, a low Tafel slope of 35 mV dec$^{-1}$, a high TOF of 3.93 s$^{-1}$ at −100 mV vs. RHE, and a mass activity about 11 times greater than the commercial Pt/C catalyst in neutral media. Besides, we reveal that the inert Co atoms are triggered by single-atom Pt, thus turning into more active sites for water dissociation under catalytic conditions, by in situ and operando XAS measurements. DFT calculations further demonstrated that the electronic interactions between atomic-level Pt and np-$Co_{0.85}Se$ can reduce energy barriers of dissociated water molecules, while significantly improve the adsorption/desorption behavior of H on Pt/np-$Co_{0.85}Se$ catalyst, which synergistically promote HER performance. This work not only provides a strategy to optimize local electronic structures of electrocatalysts for efficient hydrogen production, but also paves avenues to the further exploration and design of highly efficient electrocatalysts for other energy conversion applications.

## Methods

**Fabrication of np-$Co_{0.85}Se$.** The pure Co and CoSe powder were arc melt under an argon atmosphere to prepare Co-Se (85:15 at%) alloy ingot. Then, the alloy ingot was re-melted and rapidly quenched by the rapidly rotating copper roller to obtain thin ribbons with homogeneous nanocrystalline two-phase structure. Finally, the ribbons were selectively etched at an etching voltage of 0.0 V vs. Ag/AgCl in 0.5 M $H_2SO_4$ solution by using an electrochemical workstation (Ivium CompactStat. h). The np-$Co_{0.85}Se$ ribbons were obtained after the full etching of the Co phase (about 2000 s).

**Fabrication of Pt/np-$Co_{0.85}Se$.** Pt/np-$Co_{0.85}Se$ was synesized by a electrochemical vacancy manufacturing and atom embedding strategy, which was performed using electrochemical workstation (Ivium CompactStat. h) and a conventional three-electrode cell containing 0.5 M $H_2SO_4$ (40 ml). A np-$Co_{0.85}Se$ ribbon, a Pt foil (2 cm × 1 cm) and an Ag/AgCl electrode were used as the work electrode, counter electrode and reference electrode, respectively. CV was performed on the work electrode with a scan rate of 50 mV s$^{-1}$ between −0.2 and −0.7 V vs. Ag/AgCl.

**Characterization.** XRD measurements were conducted using a Bruker D8 Advance X-ray diffraction with Cu Kα radiation ($\lambda = 1.5418$ Å). The characterizations of morphology and elemental composition were carried out by SEM (Zeiss Sigma HD equipped with an Oxford EDS) and TEM (JEM-ARM 200F). STEM images and EDX mappings were obtained on a JEM-ARM 200F. XPS was performed on Thermo Scientific ESCALAB250Xi spectrometer equipped with an Al Kα monochromatic. The specific surface and pore diameters data were collected by using the Brunauer–Emmet–Teller (BET) and Barrett–Joyner–Halenda (BJH) methods (Micromeritics ASAP 2020), respectively. ICP-OES was performed on Agilent 730.

**X-ray absorption spectroscopy measurements.** The ex-situ X-ray absorption spectroscopy was carried out at the beamline BL01C1 in the fluorescence mode using a Lytle detector at National Synchrotron Radiation Research Center (NSRRC, Taiwan). The in-situ and operando XAS data were obtained on beamline BL01C1 at NSRRC in the fluorescence mode using a Lytle detector with a step-size of 0.25 eV at room temperature. Before the operando XAS measurements, an

electrochemical workstation (Ivium CompactStat. h) and a custom-made poly tetra fluoroethylene (PTFE) cell were used. The catalysts were coated on the carbon cloth via drop casting to form a working electrode. Then, the PTFE cell containing 1.0 M PBS was equipped with a Pt/np-Co$_{0.85}$Se working electrode, a carbon rod counter electrode, and a saturated calomel reference electrode. The fresh electrolyte was bubbled with pure argon for 1 h. Finally, the window of the PTFE cells was mounted at an angle of roughly 45° with respect to both the incident beam and the detectors. During the operando experiments, the different potentials of OCV, −0.1, and −0.2 V vs. RHE were applied to the system. The as-obtained XAS data were processed with the ATHENA program.

**Electrochemical measurements**. All electrochemical measurements were performed on an electrochemical workstation (Ivium CompactStat. h) using a three-electrode cell equipped with a graphite sheet as counter electrode and an Ag/AgCl electrode (calibrated) as reference electrode. HER measurements were conducted in 0.5 M H$_2$SO$_4$, 1.0 M KOH, 1.0 M PBS (pH = 7.0), respectively. Before test, all the fresh electrolytes were de-aerated with argon at room temperature. The sweep rate was set to 2 mV s$^{-1}$ for CV measurements. The reference electrode was converted to RHE according to the Nernst equation ($E_{RHE} = E_{Ag/AgCl} + 0.0591 \, pH + E^0_{Ag/AgCl}$). EIS spectra were performed with a frequency ranging from 10$^6$ Hz to 0.01 Hz and an amplitude of the sinusoidal voltage of 10 mV. The accelerated stability of electrodes was assessed by potential cycling between −0.4 and −0.0 V vs. RHE with a sweep rate of 100 mV s$^{-1}$. The current density–time curves were obtained at a static overpotential. To estimate the electrochemical capacitance, CV was carried out at different sweep rates. All CV curves were corrected for $iR$ losses unless noted. The calculation of current density is based on geometric area. To prepare the Pt/C electrodes, 5.0 mg Pt/C powder (10 wt%), 1 ml of isopropanol/H$_2$O (volume ratio, 1:3) mixture and 15 μl Nafion solution (5 wt%) was mixed and then ultrasonicated for about 30 min to form a homogeneous ink. After that, a certain volume of dispersion was dropped onto the glassy carbon electrode and then dried at room temperature. The average loading catalyst was ~2.04 mg cm$^{-2}$.

**Faradaic efficiency measurements**. Faradaic efficiency of Pt/np-Co$_{0.85}$Se was measured at different applied potentials (−0.01, −0.05, −0.10, −0.15, −0.20, −0.25 V vs. RHE) by Gas chromatography (GC-2014C, Shimadzu) equipped with a thermal conductivity detector for H$_2$ quantification. In a custom-made two compartment cell separated by a Nafion 117 membrane, each compartment of the cell was filled with 35 ml 1.0 M PBS solution. Ultra pure argon (Ar, 99.999%) was used as the carrier gas.

## Data availability

The data that support the plots within this paper and other findings of this study are available from the corresponding author upon reasonable request.

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

## Acknowledgements

We gratefully acknowledge financial supported by the National Natural Science Foundation of China (Grant No. 51771072), the Youth 1000 Talent Program of China, the Fundamental Research Funds for the Central Universities, and Hunan University State Key Laboratory of Advanced Design and Manufacturing for Vehicle Body Independent Research Project (No. 71860007).

## Author contributions

Y.W.T. conceived and supervised this study. K.J. and Y.Z. carried out materials fabrication, XRD/SEM/XPS characterization, and electrochemical measurements. S.C.N. conducted the TEM characterization. M.L. performed the density functional calculations and computational models. B.L., P.M., Y.R.L., T.S.C., F.M.F. de G. contributed to the XAS measurements and analysis of the XAS experiments results. Y.W.T. and K.J. wrote the paper. All authors contributed to discussions and manuscript review.

## Additional information

**Competing interests:** The authors declare no competing interests.

