## [Peer Review File · Nature Communications]

Reviewers' comments:

Reviewer #1 (Remarks to the Author):

In this manuscript, the authors reported a single-atom Pt decorated nanoporous Co_{0.85}Se (Pt/np-Co_{0.85}Se) as efficient electrocatalysts for HER. The Pt/np-Co_{0.85}Se catalyst shows excellent HER activity and durability in neutral media, outperforming commercial Pt/C catalyst. In situ and operando X-ray absorption spectroscopy studies combined with density functional theory (DFT) simulations reveals the HER thermodynamics and kinetics process. The characterizations and the mechanism analysis are clear and systematical. This work is novel and interesting. So I recommend its acceptance for publication in Nature Communications, after the following issues have been carefully addressed.

1. In Introduction Part (Page 3, line 2-4), the authors cited Ref.[15] to describe the unsatisfactory HER performance of Co_{0.85}Se at present. The Co_{0.85}Se-based catalyst reported in Ref 15 shows excellent OER and overall water splitting performance while NOT so good HER performance (particularly Tafel slope). In order to balance the recent progress about the HER performance of Co_{0.85}Se-based catalysts, the authors should cite several references about Co_{0.85}Se-based catalyst with better HER performance in alkaline solution (e.g. J Power Sources, 400, 232, 2018 and J. Mater. Chem. A, 2017, 5, 7001) and in acidic solution (e.g. Electrochimica Acta, 247, 468 and ACS Appl Mater Inter, 9, 30703).
2. As the XRD patterns (Figure. 1b) show that a shift was caused by single-atom Pt doping. I think the influence of Pt doping on the substrate should be further discussed.
3. The FT-EXAFS results at Pt L₃-edge show a conspicuous peak at 2.03 Å from the Pt-Se (as the authors mention) contribution (Figure. 2c & page 7. Line 13). How to exclude the possibility of Pt-Co contribution?
4. The authors are recommended to provide longer chronoamperometry data (Figure. 3j) to demonstrate long-term stability of their catalysts.
5. Pt sites on Pt/Co_{0.85}Se should play an important role in Tafel step. But the authors did not add Pt sites to the comparison in Free energy diagrams for hydrogen adsorption (Figure. 5d). It would be more valuable if the authors could take this into account.
6. In the Supplementary Note 1 section, the authors mention the mechanism of Co vacancies formation. I think the author needs to cite literatures for further explanation.

Reviewer #2 (Remarks to the Author):

This manuscript (Manuscript Number: NCOMMS-19-01442) reports how platinum atoms can be embedded in nanoporous cobalt selenide to enhance the electrocatalytic

activity toward HER in neutral media. The electrocatalyst sample has been well characterized using a wide range of characterization techniques and has been well evaluated as a HER electrocatalyst. However, the manuscript has lack of novelty (for publishing in a very high-rank journal like Nature Communications) and there is not a good literature review in the introduction and discussion sections. Employing the potential-cycling synthesis method to produce atomic-scale Pt as active electrocatalysts toward HER is not any more a new idea (there are many published papers in this field in recent years, which have not been mentioned in this manuscript) and I think this manuscript does not have a high level of novelty required for Nature Communications.

The potential-cycling synthesis method in this manuscript has been previously presented as a facile method to decorate atomic scale Pt on various support materials such as carbon nanotubes (ACS Catal. 2017, 7, 5, 3121-3130, DOI: 10.1021/acscatal.7b00199), CoP nanotube arrays (Angew. Chem. Int. Ed., 2017, 56, 13694-13698, DOI: 10.1002/anie.201706921), etc. to make active and stable HER electrocatalysts. Actually, using this synthesis method we can activate a wide range of materials for HER. It is shown before that atomic scale Pt containing ultra-low amount of Pt can show extremely higher mass activity in comparison to that of Pt/C (see for example above-mentioned references). The authors had to cite at least the papers that could make active atomic scale Pt (as active HER electrocatalysts) using the same synthesis method but unfortunately, they have totally ignored such relative publications. The authors have applied 5000 activation cycles to activate the CoSe support in 0.5 M H₂SO₄ which shows a low activation rate with Pt for this support, in comparison to that for other reported supports. The activity toward neutral HER is also lower than, for example, CoP nanotube support which is activated with atomic Pt using a similar synthesis method (see Angew. Chem. Int. Ed., 2017, 56, 13694-13698, DOI: 10.1002/anie.201706921).

In case the authors would like to improve their manuscript, the following comments may be also helpful.

1. The authors have not mentioned any of such previously published papers which have used a similar synthesis method to deposit atomic scale Pt as active HER electrocatalyst and instead they have compared the electrocatalytic performance of the Pt/np-Co_{0.85}Se with Pt/C and transition metal-based catalysts (Table S3). Improve the literature review and cite the papers with more relevant contents (atomic scale Pt for HER) as explained above. Note specifically what makes this manuscript different and important in comparison to those papers.

2. The cyclic voltammograms (CV) of the Pt/np-Co_{0.85}Se (in comparison to np-Co_{0.85}Se and Pt/C) can be highly informative for electrochemists. For atomic scale Pt, the CV features are different to that of the typical CV of bulk, nanostructure, and clusters of Pt (see ACS Catal. 2017, 7, 8033–8041, DOI: 10.1021/acscatal.7b02878). I

recommend measuring CVs in the range of 0-1.2 V vs. RHE (in the buffer solution and 0.5 M H₂SO₄) to see the whole Pt CV features to predict/confirm the structure of Pt. Cyclic voltammogram is a very simple and fast measurement that can be very informative.

3. In this manuscript, bulk Pt foil has been used as a reference for Pt⁰ in XPS measurements, which is not a good reference. I recommend selecting a sample containing small Pt NPs as the reference for Pt⁰. For instance, look at this reference (Angew. Chem. Int. Ed. 2017, 56, 13694-13698, DOI: 10.1002/anie.201706921) in which commercial Pt/C containing Pt nanoparticles displays two Pt 4f peaks at 71.6 and 74.9 eV, indicative of Pt⁰. These values are close to the values for Pt 4f in this manuscript and it makes it more difficult to prove if the Pt is in the ionic state. Thus, based on XPS results, it is not well clear if Pt is metallic or ionic Pt.

Reviewer #3 (Remarks to the Author):

In this manuscript, the authors report a single-atom Pt decorated nanoporous Co_{0.85}Se (Pt/np-Co_{0.85}Se) catalyst, which has the ability to efficiently and robustly catalyze the hydrogen evolution reaction (HER). The Pt/np-Co_{0.85}Se catalyst with an ultralow single-atom Pt loading while possess a near-zero onset overpotential, a high turnover frequency, a high mass activity, and a low Tafel slope, outperforming commercial Pt/C catalyst and other reported transition-metal-based compounds in neutral electrolyte. In addition, Pt/np-Co_{0.85}Se catalyst has also highly active and stability toward HER in acidic and basic electrolytes.

More importantly, the HER enhancement mechanism of Pt doping is demonstrated by In situ and operando X-ray absorption spectroscopy studies and density functional theory (DFT) simulations. The presence of single-atom Pt in Pt/np-Co_{0.85}Se not only can optimize surface states of Co_{0.85}Se active centers and reduce energy barriers of dissociated water molecules, but also can significantly improve adsorption/desorption behavior of hydrogen, which synergistically promote HER thermodynamics and kinetics. It is impressive that the Pt doping could trigger the activity of Co_{0.85}Se by effectively optimize the d-electron domination of Co atoms, thus leading to enhanced catalytic activity.

I think that this work is well done, and the results and explanations are all convincing. This work is the first to explore that how single-atom metal is beneficial to promote HER activity of current electrocatalysts, especially for cobalt selenides, and thus this manuscript will attract much attention from readers working on energy science and systems, materials science and catalysis. It should be published on Nature Communications, only after the following minor issues are addressed:

1. The inset of Figure 2b shows that the average oxidation state of Pt in Pt/np-Co_{0.85}Se is 0.8. But the authors did not mention it in this manuscript.
2. The labeling of Co²⁺ and Co³⁺ in Figure 4c and d is unnecessary because the authors did not make any further explanation.

3. In DFT simulations, the authors used both Pt/np-Co_{0.85}Se (Page 12-13) and Pt/Co_{0.85}Se (Page 13 and Figure 5) to represent catalyst. I think the authors should use Pt/Co_{0.85}Se as a unified representation in this part.

4. In the HER mechanism, the generated H_{ads} could be adsorbed on a nearby empty Co site or Pt site and further be converted into H₂ readily through the Tafel step (Page 12, line 6-8). However, the free energy of Pt sites for hydrogen adsorption was not calculated in the diagram (Figure 5d).

5. Supplementary Note 3: I think the calculation formula of TOF values of np-Co_{0.85}Se also should be listed here.

Response to Reviewers' Comments

Reviewer #1:

In this manuscript, the authors reported a single-atom Pt decorated nanoporous Co_{0.85}Se (Pt/np-Co_{0.85}Se) as efficient electrocatalysts for HER. The Pt/np-Co_{0.85}Se catalyst shows excellent HER activity and durability in neutral media, outperforming commercial Pt/C catalyst. In situ and operando X-ray absorption spectroscopy studies combined with density functional theory (DFT) simulations reveals the HER thermodynamics and kinetics process. The characterizations and the mechanism analysis are clear and systematical. This work is novel and interesting. So I recommend its acceptance for publication in Nature Communications, after the following issues have been carefully addressed.

Reply: We appreciate the reviewer for his/her constructive comments and for supporting our paper for publication in Nature Communications. Following the comments and suggestions, we have conducted additional experiments and analyses and carefully revised the manuscript. The details will be described below.

1. In Introduction Part (Page 3, line 2-4), the authors cited Ref.[15] to describe the unsatisfactory HER performance of Co_{0.85}Se at present. The Co_{0.85}Se-based catalyst reported in Ref 15 shows excellent OER and overall water splitting performance while NOT so good HER performance (particularly Tafel slope). In order to balance the recent progress about the HER performance of Co_{0.85}Se-based catalysts, the authors should cite several references about Co_{0.85}Se-based catalyst with better HER performance in alkaline solution (e.g. J Power Sources, 400, 232, 2018 and J. Mater. Chem. A, 2017, 5, 7001) and in acidic solution (e.g. Electrochimica Acta, 247, 468 and ACS Appl Mater Inter, 9, 30703).

Reply: All the references suggested by the reviewer have been cited in the revised manuscript (Page 3, line 3). Thanks.

2. As the XRD patterns (Figure. 1b) show that a shift was caused by single-atom Pt

doping. I think the influence of Pt doping on the substrate should be further discussed.

Reply: We sincerely appreciate your valuable suggestion. We added the XRD patterns in **supplementary Figure 9** of the revised supplementary information. The relevant discussion of the XRD patterns is shown in revised manuscript (Page 5, line 13-16).

As Figure R1 shown, X-ray diffraction (XRD) of Pt/np-Co_{0.85}Se shows the characteristic peaks of a hexagonal structure with the lattice parameters nearly identical to those of Co_{0.85}Se (JPCDS card no. 52-1008) excepting for a slight shift, indicating that the atomic-level Pt dopant did not form crystalline segregated bi-phases, but was well-incorporated into the Co_{0.85}Se crystal lattice.

Figure R1. XRD results at (102) and (110) peaks of np-Co_{0.85}Se and Pt/np-Co_{0.85}Se.

3. The FT-EXAFS results at Pt L₃-edge show a conspicuous peak at 2.03 Å from the Pt-Se (as the authors mention) contribution (Figure. 2c & page 7. Line 13). How to exclude the possibility of Pt-Co contribution?

Reply: We thank the reviewer for the insightful comment. The FT-EXAFS spectra (**Fig. 2c**) show a conspicuous peak at a radial distance of 2.03 Å which is much shorter than that of Pt-Co shell (2.55 Å) (*J. Catal.* 2008, 259, 260-268, DOI: 10.1016/j.jcat.2008.08.016). Therefore, this conspicuous peak should correspond with Pt-Se shell rather than Pt-Co shell. Furthermore, as **Figure 2e** shown, the enhanced intensity of Se-Co (Se-Pt) shell in the FT-EXAFS signals indicates the more coordination number of Se atoms, which also proves the existence of Pt-Se shell rather than Pt-Co shell.

4. The authors are recommended to provide longer chronoamperometry data (Figure. 3j) to demonstrate long-term stability of their catalysts.

Reply: Thanks for your suggestion. The longer chronoamperometry data are presented in **Figure. 3j** of the revised manuscript.

5. Pt sites on Pt/Co_{0.85}Se should play an important role in Tafel step. But the authors did not add Pt sites to the comparison in Free energy diagrams for hydrogen adsorption (Figure. 5d). It would be more valuable if the authors could take this into account.

Reply: We thank the reviewer for the constructive comment. Following this comment, we calculated the hydrogen adsorption free energy of the Pt sites on Pt/Co_{0.85}Se. Please see **Figure 5d** of the revised manuscript. We also added the discussion for the Pt sites in the revised manuscript (Page 13, line 15-18).

6. In the Supplementary Note 1 section, the authors mention the mechanism of Co vacancies formation. I think the author needs to cite literatures for further explanation.

Reply: We thank the reviewer for the insightful comments. Following the comments, we cited correlative references about the formation of Co vacancies (Ref. 1 of the revised supplementary information: *Nat. Mater.* 2012, 11, 775-780). To further clarify the formation of Co vacancies, we also conducted Fourier transform EXAFS spectra at the Se K-edge of the np-Co_{0.85}Se and the np-Co_{0.85}Se obtained by a 5000 cyclic voltammetry using graphite sheet as counter electrode in a three-electrode cell containing 0.5 M H₂SO₄ to get more information on their different atomic structure. The analysis results are shown in **Supplementary Note 1** and **Supplementary Figure 6** of the revised supplementary information. As Figure R2 shown, after 5000 cyclic voltammetry using graphite sheet as counter electrode in a three-electrode cell containing 0.5 M H₂SO₄, the Se-Co shell scattering was decreased, indicating the loss of Co atoms, thus providing anchor sites for Pt atoms. (*Nat. Commun.* 2018, **9**, 1002).

Figure R2. Fourier transform of the EXAFS spectra in real space at Se K-edge. Blue curve: np-Co_{0.85}Se. Red curve: np-Co_{0.85}Se obtained by a 5000 cyclic voltammetry using graphite sheet as counter electrode in a three-electrode cell containing 0.5 M H₂SO₄.

Reviewer #2:

This manuscript (Manuscript Number: NCOMMS-19-01442) reports how platinum

atoms can be embedded in nanoporous cobalt selenide to enhance the electrocatalytic activity toward HER in neutral media. The electrocatalyst sample has been well characterized using a wide range of characterization techniques and has been well evaluated as a HER electrocatalyst. However, the manuscript has lack of novelty (for publishing in a very high-rank journal like Nature Communications) and there is not a good literature review in the introduction and discussion sections. Employing the potential-cycling synthesis method to produce atomic-scale Pt as active electrocatalysts toward HER is not any more a new idea (there are many published papers in this field in recent years, which have not been mentioned in this manuscript) and I think this manuscript does not have a high level of novelty required for Nature Communications.

The potential-cycling synthesis method in this manuscript has been previously presented as a facile method to decorate atomic scale Pt on various support materials such as carbon nanotubes (*ACS Catal.* 2017, 7, 5, 3121-3130, DOI: 10.1021/acscatal.7b00199), CoP nanotube arrays (*Angew. Chem. Int. Ed.*, 2017, 56, 13694-13698, DOI: 10.1002/anie.201706921), etc. to make active and stable HER electrocatalysts. Actually, using this synthesis method we can activate a wide range of materials for HER. It is shown before that atomic scale Pt containing ultra-low amount of Pt can show extremely higher mass activity in comparison to that of Pt/C (see for example above-mentioned references). The authors had to cite at least the papers that could make active atomic scale Pt (as active HER electrocatalysts) using the same synthesis method but unfortunately, they have totally ignored such relative publications. The authors have applied 5000 activation cycles to activate the CoSe support in 0.5 M H₂SO₄ which shows a low activation rate with Pt for this support, in comparison to that for other reported supports. The activity toward neutral HER is also lower than, for example, CoP nanotube support which is activated with atomic Pt using a similar synthesis method (see *Angew. Chem. Int. Ed.*, 2017, 56, 13694-13698, DOI: 10.1002/anie.201706921).

Reply: We appreciate the reviewer for recognizing the importance of our finding of the free-standing nanoporous Pt/Co_{0.85}Se catalyst and for his/her constructive comments, which have been addressed point-by-point below. We also thank the reviewer for bringing out our attention to the previous important work (*ACS Catal.* 2017, 7, 5, 3121-3130; *Angew. Chem. Int. Ed.*, 2017, 56, 13694-13698) on potential-cycling synthesis method. Here, however, we will firstly clarify the main contribution of this work, then we will point out differences between this work and the mentioned reports (*ACS Catal.* 2017, 7, 5, 3121-3130; *Angew. Chem. Int. Ed.*, 2017, 56, 13694-13698). In our opinion, those works should not affect the importance and novelty of the current report for following reasons:

- 1). We agree with the reviewer's point that employing the potential-cycling synthesis method to produce atomic-scale Pt as active electrocatalysts toward HER has been reported in previous work. We feel sorry for neglecting those literatures with relevant

contents (*ACS Catal.*, 2017, 7, 5, 3121-3130; *Angew. Chem. Int. Ed.*, 2017, 56, 13694-13698). Based on your suggestions, we add those literatures to the revised manuscript (please see **Comment I.**). Generally, fabrication of single-atom catalyst is a major challenge because of the tendency of single metal atom aggregation. Laasonen and co-workers firstly reported a potential cycling method to deposit atomic scale Pt on single-walled carbon nanotubes as active HER electrocatalyst in 2017 (*ACS Catal.* 2017, 7, 5, 3121-3130). This method of depositing ultralow Pt species on working electrodes by CV is ingenious and it can avoid the formation of Pt nanoparticles. However, the role of CV on the substrate (working electrode) was not further explored. In our work, CV acts as an alternating redox process for Pt species, which is indeed the same as previously reported literatures (*ACS Catal.* 2017, 7, 5, 3121-3130; *Angew. Chem. Int. Ed.*, 2017, 56, 13694-13698). But in our fabrication method, CV is more importantly used to create Co vacancies on Co_{0.85}Se which acts as the anchoring sites for single Pt atoms. This is key to obtain the single atom catalyst, because all the time the fabrication of stable single-atom catalysts remains a formidable challenge due to the difficulty of creating high densities of underpinning stable defects (*Joule*, 2018, 2, 1242-1264). Thus, our fabrication method provides a new method for vacancy manufacturing with vacancy manufacturing and Pt atom anchoring occurring simultaneously. The success of single-atom catalysts preparation requires that the rate of vacancy generation is faster than that of Pt atom anchoring. When the scan rate exceeds 50 mV s⁻¹, Pt nanoparticles are easily formed. This is because the faster scan rate accelerates the dissolution of Pt species on counter electrode, resulting in the faster rate of Pt atom anchoring (*ACS Energy Lett.* 2017, 2, 1070-1075). To clearly highlight the contribution of this work, relevant experimental data and corresponding discussions were added in the revised manuscript (Page 4, line 19-22; Page 5, line 1-3) and the supplementary information (**Supplementary Note 1**). We believe that fabricating stable defects by CV is an important method which could draw more attention of researchers to the potential cycling method. For your convenience, all changes made for the responses have been highlighted by yellow in the revised manuscript and the supplementary information files.

2) After carefully compared to the reviewer mentioned the paper (*Angew. Chem. Int. Ed.*, 2017, 56, 13694-13698). Although the overpotential at 10 mA cm⁻² of Pt/np-Co_{0.85}Se is slightly higher than that of PtSA-NT-NF (*Angew. Chem. Int. Ed.*, 2017, 56, 13694-13698), there are many merits (intrinsic catalysis activities) of Pt/np-Co_{0.85}Se are comparable with that of PtSA-NT-NF. For instance, Pt/np-Co_{0.85}Se catalyst shows higher mass activity than that of PtSA-NT-NF (Pt/np-Co_{0.85}Se: 1.32 A mg⁻¹, PtSA-NT-NF: 0.36 A mg⁻¹) (see Table R1 and **Supplementary Table 1** of the revised supplementary information.)

3) Although single-atom catalyst has been extensively studied for HER (*Sci. Adv.* 2018, 4, eaao6657; *Angew. Chem. Int. Ed.*, 2017, 56, 13694-13698; *ACS Energy Lett.* 2018, 3, 1713-1721), up to now, there is still no conclusion about the enhancing mechanism of single-atom doping for current HER electrocatalysts during the HER.

Therefore, we would also like to emphasize that one of the primary motivations of this work is to demonstrate how single-atom metal is beneficial to promote HER activity of current electrocatalysts under realistic reaction conditions, especially for cobalt selenides. Fortunately, we demonstrated the mechanism according to the change of signals during HER by using in situ and operando X-ray absorption spectroscopy results and together with theoretical support. Hence, this work offers good guidance for design and fabrication of electrocatalysts to effectively manipulate its catalytic properties by an atomic-level engineering strategy.

Considering the novelty, scientific importance and potential applications, we wish the reviewer can share our enthusiasms that this work indeed represents one of the most important findings in the current development of single-atom catalysts enhancing HER mechanism and it deserves to be published in *Nature Communications*.

In case the authors would like to improve their manuscript, the following comments may be also helpful.

1. The authors have not mentioned any of such previously published papers which have used a similar synthesis method to deposit atomic scale Pt as active HER electrocatalyst and instead they have compared the electrocatalytic performance of the Pt/np-Co_{0.85}Se with Pt/C and transition metal-based catalysts (Table S3). Improve the literature review and cite the papers with more relevant contents (atomic scale Pt for HER) as explained above. Note specifically what makes this manuscript different and important in comparison to those papers.

Reply: Thank you for your helpful suggestions. As you suggested, description of correlative work about the deposit atomic scale Pt and its inspiration to us have been added into the revised manuscript (Page 3, line 10-13; Page 4, line 19-20). Corresponding literatures (Ref. 33: *ACS Catal.* 2017, 7, 5, 3121-3130; Ref. 32: *Angew. Chem. Int. Ed.* 2017, 56, 13694-13698) also have been cited in the revised manuscript. In addition, relevant experimental data and corresponding discussion were added in the revised manuscript (Page 4, line 19-22; Page 5, line 1-3) and supplementary information (**Supplementary Note 1 and Fig. 6**) to show the special features of our work. In order to compare the HER performance of Pt/np-Co_{0.85}Se and previously reported Pt-based catalysts (include catalysts prepared by CV), **Table 1** was also added in revised supplementary information.

Table R1. Comparison of overpotential (η) at current density of -10 mA cm^{-2} , Tafel slopes and mass activity of Pt/np-Co_{0.85}Se with recently reported Pt-based catalysts.

Catalysts with electrodepositing atomic scale Pt.

Catalysts	electrolyte	η_{10} (mV)	Tafel slope (mV dec ⁻¹)	Mass activity at $\eta=100$ mV (A mg ⁻¹)	Ref.
Pt/np-Co _{0.85} Se	1.0 M PBS	55	35	1.32	This work
Pt/np-Co _{0.85} Se	0.5 M H ₂ SO ₄	58	26	13.57	This work
Pt/np-Co _{0.85} Se	1.0 M KOH	58	39	1.28	This work
Pt/C	1.0 M PBS	46	37	0.12	This work
Pt/C	0.5 M H ₂ SO ₄	29	31	0.78	This work
Pt/C	1.0 M KOH	40	46	0.20	This work
#400-SWNT/Pt	0.5 M H ₂ SO ₄	27	38	About 3.30	ACS Catal. 7 , 3121-3130 (2017).
#PtSA-NT-NF	1.0 M PBS	24	30	About 0.36	Angew. Chem. Int. Ed. 56 , 13694-13698 (2017).
#PtSA-NT-NF	0.5 M H ₂ SO ₄	30	-	About 0.93	Angew. Chem. Int. Ed. 56 , 13694-13698 (2017).
#PtSA-NT-NF	1.0 M KOH	20	-	About 0.54	Angew. Chem. Int. Ed. 56 , 13694-13698 (2017).
#er-WS ₂ -Pt	0.5 M H ₂ SO ₄	About 45	27	-	Adv. mater. 30 , 1704779(2018).
#er-WS ₂ -Pt	1.0 M KOH	About 48	65	-	Adv. mater. 30 , 1704779(2018).
#ep-WS ₂ -Pt	0.5 M H ₂ SO ₄	About 140	50	-	Adv. mater. 30 , 1704779(2018).
#ep-WS ₂ -Pt	1.0 M KOH	About 190	124	-	Adv. mater. 30 , 1704779(2018).
#Pt-2H-MoS ₂	0.5 M H ₂ SO ₄	312	109	-	Chem. Mater. 31 , 429-435(2019).
#Pt-1T-MoS ₂	0.5 M H ₂ SO ₄	210	104	-	Chem. Mater. 31 , 429-435(2019).

Pt@PCM	0.5 M H ₂ SO ₄	105	65.3	About 0.10	Sci. Adv. 4 , eaao6657 (2018)
Pt@PCM	1.0 M KOH	139	73.6	-	Sci. Adv. 4 , eaao6657 (2018)
Pt@MoS ₂ /NiS ₂	0.5 M H ₂ SO ₄	34	41	About 7.30	Small 14 , 1800679(2018)
Pt-MoS ₂	0.5 M H ₂ SO ₄	53	40	About 1.74	Nat. Commun. 4 , 1444 (2013).
Pt-Co(OH) ₂ /CC	1.0 M PBS	84	-	About 0.03	ACS Catal. 7 , 7131–7135 (2017).

2. The cyclic voltammograms (CV) of the Pt/np-Co_{0.85}Se (in comparison to np-Co_{0.85}Se and Pt/C) can be highly informative for electrochemists. For atomic scale Pt, the CV features are different to that of the typical CV of bulk, nanostructure, and clusters of Pt (see *ACS Catal.* 2017, 7, 8033–8041, DOI: 10.1021/acscatal.7b02878). I recommend measuring CVs in the range of 0–1.2 V vs. RHE (in the buffer solution and 0.5 M H₂SO₄) to see the whole Pt CV features to predict/confirm the structure of Pt. Cyclic voltammogram is a very simple and fast measurement that can be very informative.

Reply: We appreciate you for this insightful and constructive recommendation on the structure determination of our Pt/np-Co_{0.85}Se material. Following this suggestion, we performed the CV measurement for np-Co_{0.85}Se, Pt/np-Co_{0.85}Se, and Pt/C in N₂ purged 0.5 M H₂SO₄ and 1.0 M PBS with a scan rate of 50 mV s⁻¹ (See Figure R3). We added the new data in **Supplementary Figure 8** of the revised supplementary information and a brief discussion in revised manuscript (Page 5, line 7–11) to predict/confirm the structure of Pt.

As Figure R3 shown, CV measurements under non-catalytic conditions were used to detect the feature voltammetric response of Pt (Ref. 39: *ACS Catal.* 2017, 7, 8033–8041). Regions of interest include the Pt-H adsorption/desorption peaks and Pt-O formation/reduction peak. However, these typical features for Pt cannot be observed on Pt/np-Co_{0.85}Se, indicative of ultralow Pt loading.

Figure R3. Cyclic voltammograms for np-Co_{0.85}Se, Pt/np-Co_{0.85}Se, and Pt/C in N₂

purged (a) 0.5 M H₂SO₄ and (b) 1.0 M PBS with a scan rate of 50 mV s⁻¹.

3. In this manuscript, bulk Pt foil has been used as a reference for Pt⁰ in XPS measurements, which is not a good reference. I recommend selecting a sample containing small Pt NPs as the reference for Pt⁰. For instance, look at this reference (Angew. Chem. Int. Ed. 2017, 56, 13694-13698, DOI: 10.1002/anie.201706921) in which commercial Pt/C containing Pt nanoparticles displays two Pt 4f peaks at 71.6 and 74.9 eV, indicative of Pt⁰. These values are close to the values for Pt 4f in this manuscript and it makes it more difficult to prove if the Pt is in the ionic state. Thus, based on XPS results, it is not well clear if Pt is metallic or ionic Pt.

Reply: We thank for the reviewer's valuable suggestions. Following the suggestions, we measured the XPS spectral of commercial Pt/C. The relevant data are added in **Figure 2a** of the revised manuscript. As Figure R4 shown, the Pt 4f spectrum of Pt/np-Co_{0.85}Se suggests that the binding energy of Pt (71.8 eV) is close to that of Pt⁰ (71.2 eV) with a slightly positive shift, indicating Pt atoms carry partially positive charge through electron transfer between metal and supports.

Figure R4. XPS spectra of Pt/np-Co_{0.85}Se and commercial Pt/C in Pt 4f regions.

Reviewer #3:

In this manuscript, the authors report a single-atom Pt decorated nanoporous Co_{0.85}Se (Pt/np-Co_{0.85}Se) catalyst, which has the ability to efficiently and robustly catalyze the hydrogen evolution reaction (HER). The Pt/np-Co_{0.85}Se catalyst with an ultralow single-atom Pt loading while possess a near-zero onset overpotential, a high turnover frequency, a high mass activity, and a low Tafel slope, outperforming commercial Pt/C catalyst and other reported transition-metal-based compounds in neutral electrolyte. In addition, Pt/np-Co_{0.85}Se catalyst has also highly active and stability toward HER in acidic and basic electrolytes.

More importantly, the HER enhancement mechanism of Pt doping is demonstrated by In situ and operando X-ray absorption spectroscopy studies and density functional theory (DFT) simulations. The presence of single-atom Pt in Pt/np-Co_{0.85}Se not only can optimize surface states of Co_{0.85}Se active centers and reduce energy barriers of dissociated water molecules, but also can significantly improve adsorption/desorption behavior of hydrogen, which synergistically promote HER thermodynamics and

kinetics. It is impressive that the Pt doping could trigger the activity of $\text{Co}_{0.85}\text{Se}$ by effectively optimize the d-electron domination of Co atoms, thus leading to enhanced catalytic activity.

I think that this work is well done, and the results and explanations are all convincing. This work is the first to explore that how single-atom metal is beneficial to promote HER activity of current electrocatalysts, especially for cobalt selenides, and thus this manuscript will attract much attention from readers working on energy science and systems, materials science and catalysis. It should be published on Nature Communications, only after the following minor issues are addressed:

Reply: We thank the reviewer for his/her valuable comments/suggestions and for pointing out the novelty and importance of our work. Following the comments, we have carefully revised our manuscript and the details are listed below.

1. The inset of Figure 2b shows that the average oxidation state of Pt in Pt/np- $\text{Co}_{0.85}\text{Se}$ is 0.8. But the authors did not mention it in this manuscript.

Reply: We thank the reviewer for the insightful comments. Following this comment, we added the discussion for describe the inset of **Figure 2b** in the revised manuscript (Page 7, Line 17).

2. The labeling of Co^{2+} and Co^{3+} in Figure 4c and d is unnecessary because the authors did not make any further explanation.

Reply: Thank you for this comment. We have deleted the labeling of Co^{2+} and Co^{3+} in **Figure 4c** and **d**.

3. In DFT simulations, the authors used both Pt/np- $\text{Co}_{0.85}\text{Se}$ (Page 12-13) and Pt/ $\text{Co}_{0.85}\text{Se}$ (Page 13 and Figure 5) to represent catalyst. I think the authors should use Pt/ $\text{Co}_{0.85}\text{Se}$ as a unified representation in this part.

Reply: We thank the reviewer for careful reading our paper. We have corrected typos in the revised manuscript.

4. In the HER mechanism, the generated H_{ads} could be adsorbed on a nearby empty Co site or Pt site and further be converted into H_2 readily through the Tafel step (Page 12, line 6-8). However, the free energy of Pt sites for hydrogen adsorption was not calculated in the diagram (Figure 5d).

Reply: We thank the reviewer for the constructive comment. Following this comment, we calculated the hydrogen adsorption free energy of the Pt sites on Pt/ $\text{Co}_{0.85}\text{Se}$. Please see **Figure 5d** of the revised manuscript. We also added the discussion for the Pt sites in revised manuscript (Page 13, line 15-18).

5. Supplementary Note 3: I think the calculation formula of TOF values of np-Co_{0.85}Se also should be listed here.

Reply: We appreciate you for this valuable suggestion. Following your comment, the calculation formula of TOF values of np-Co_{0.85}Se are presented in the **Supplementary Note 3** of the revised supplementary information as follows:

The active site density of Co and Se is estimated by using the method suggested by Ref. 5 (*Energ. Environ. Sci.* 2015, 8, 3022-3029):

$$\begin{aligned} \text{np-Co}_{0.85}\text{Se: } & 9.021 \times 10^{14} \text{ atoms cm}_{\text{real}}^{-2} \times (\text{BET}_2) \times (\text{mass}) \\ & = 2.406 \times 10^{16} \text{ Co and Se - sites cm}^{-2} \end{aligned}$$

Finally, the current density from the LSV polarization curve can be converted into TOF values according to:

$$\text{TOF}_{\text{np-Co}_{0.85}\text{Se}} = \frac{3.12 \times 10^{15} \text{ H}_2 \text{ s}^{-1} \text{ cm}^{-2} \text{ per mA cm}^{-2}}{2.406 \times 10^{16} \text{ Co and Se - sites cm}^{-2}} \times j = 0.129 \text{ j}$$

Reviewers' comments:

Reviewer #1 (Remarks to the Author):

The authors have addressed all issues raised by the referees and it can be accepted in the present form.

Reviewer #2 (Remarks to the Author):

The authors have well addressed the reviewer's comments and revised the manuscript accordingly and thus it can be published on Nature Communications. I only noticed that the following added comment could be clarified better by the authors.

“However, at present, there is no conclusion how single-atom metal is beneficial to promote HER activity of current electrocatalysts under realistic reaction conditions, ...”

I think it is been well studied both computationally and experimentally so far how the Pt single-atom is beneficial to promote HER activity especially for acidic and neutral conditions (look at the newly added references for instance). Thus, the above-mentioned comment in the manuscript would be only valid for the cobalt selenide as the support. However, in comparison to other previously reported atomic Pt catalysts for HER, this work investigates comprehensively an atomic Pt catalyst in all media from acidic to neutral and alkaline conditions. As far as I know, the previously reported atomic scale Pt catalysts have not considered the activity in all media either because of the lack of high HER reactivity in all media or just because

those catalysts have not been also evaluated in other media.

Reviewer #3 (Remarks to the Author):

I read it carefully and found that the authors have made a commendable effort to address all the concerns of mine and the other reviewers'. Therefore, I am satisfied with the revised version of the paper. I believe that this paper is publishable in Nature Communications.

Response to Reviewers' Comments

Reviewer #1:

The authors have addressed all issues raised by the referees and it can be accepted in the present form.

Reply: We appreciate your recommendation of acceptance and helpful comments in the reviewing process and are pleased to have our manuscript be reviewed by you.

Reviewer #2:

The authors have well addressed the reviewer's comments and revised the manuscript accordingly and thus it can be published on Nature Communications. I only noticed that the following added comment could be clarified better by the authors.

“However, at present, there is no conclusion how single-atom metal is beneficial to promote HER activity of current electrocatalysts under realistic reaction conditions, ...”

I think it is been well studied both computationally and experimentally so far how the Pt single-atom is beneficial to promote HER activity especially for acidic and neutral conditions (look at the newly added references for instance). Thus, the above-mentioned comment in the manuscript would be only valid for the cobalt selenide as the support. However, in comparison to other previously reported atomic Pt catalysts for HER, this work investigates comprehensively an atomic Pt catalyst in all media from acidic to neutral and alkaline conditions. As far as I know, the previously reported atomic scale Pt catalysts have not considered the activity in all media either because of the lack of high HER reactivity in all media or just because those catalysts have not been also evaluated in other media.

Reply: We are very grateful to your encouraging and positive comments and really appreciate your agreement of acceptance with this revised manuscript. As you suggested, we replaced the relevant description with the following sentence:

However, at present, direct insights into how single-atom metal is beneficial to

promote HER activity of current electrocatalysts are rarely available under realistic reaction conditions, especially for cobalt selenides.

Reviewer #3:

I read it carefully and found that the authors have made a commendable effort to address all the concerns of mine and the other reviewers'. Therefore, I am satisfied with the revised version of the paper. I believe that this paper is publishable in Nature Communications.

Reply: We truly thank you for reviewing the revised version of our manuscript and greatly appreciate your helpful and affirmative comments.